# VEGF Induces Expression of Genes That Either Promote or Limit Relaxation of the Retinal Endothelial Barrier

**DOI:** 10.3390/ijms24076402

**Published:** 2023-03-29

**Authors:** Maximilian McCann, Yueru Li, Basma Baccouche, Andrius Kazlauskas

**Affiliations:** 1Department of Ophthalmology & Visual Sciences, University of Illinois at Chicago, Chicago, IL 60612, USA; 2Department of Physiology & Biophysics, University of Illinois at Chicago, Chicago, IL 60612, USA

**Keywords:** VEGF, Notch pathway, Ang2

## Abstract

The purpose of this study was to identify genes that mediate VEGF-induced permeability. We performed RNA-Seq analysis on primary human retinal endothelial cells (HRECs) cultured in normal (5 mM) and high glucose (30 mM) conditions that were treated with vehicle, VEGF, or VEGF then anti-VEGF. We filtered our RNA-Seq dataset to identify genes with the following four characteristics: (1) regulated by VEGF, (2) VEGF regulation reversed by anti-VEGF, (3) regulated by VEGF in both normal and high glucose conditions, and (4) known contribution to vascular homeostasis. Of the resultant 18 genes, members of the Notch signaling pathway and *ANGPT2* (Ang2) were selected for further study. Permeability assays revealed that while the Notch pathway was dispensable for relaxing the barrier, it contributed to maintaining an open barrier. In contrast, Ang2 limited the extent of barrier relaxation in response to VEGF. These findings indicate that VEGF engages distinct sets of genes to induce and sustain barrier relaxation. Furthermore, VEGF induces expression of genes that limit the extent of barrier relaxation. Together, these observations begin to elucidate the elegance of VEGF-mediated transcriptional regulation of permeability.

## 1. Introduction

The integrity of the paracellular barrier between endothelial cells of the retinal vasculature is crucial for proper retinal function. A compromised endothelial barrier leads to vessel leakage and the development of edema and visual impairments in patients with retinopathies like diabetic retinopathy (DR) and neovascular age-related macular degeneration (nAMD) [1,2]. There are various secreted factors that compromise the retinal endothelial barrier and are elevated in patients with retinopathies [3,4,5]. Vascular endothelial growth factor (VEGF) is one of the most widely studied, and the target of several neutralizing therapies that improve the vision of patients with retinopathies [6]. While anti-VEGF therapies have proven to be effective at improving visual acuity in patients with both DR and nAMD, anti-VEGF is not effective for all patients and efficacy can wane over time [7]. Therefore, alternative therapies are needed, and their development will be guided by a better understanding of how VEGF disrupts the endothelial barrier.

VEGF compromises the endothelial barrier in two stages. In the first stage, it interacts with the receptor tyrosine kinase (RTK) vascular endothelial growth factor receptor 2 (VEGFR2) to induce signaling events that weaken adherens junctions and cause endothelial permeability [8,9]. The second stage consists of a prolonged disruption of the barrier that has not been fully described. Its slow onset suggests that VEGF induces transcriptional changes that both initiate and sustain this barrier disruption. Our lab previously identified genes that are induced by VEGF and are necessary for VEGF to maximally open the endothelial barrier formed by primary human retinal endothelial cells (HRECs). VEGF increases expression of angiotensin-converting enzyme (*ACE*), and this change is reversed by anti-VEGF [10]. Inhibiting ACE or blocking the angiotensin II receptor prevents maximal relaxation of the barrier in response to VEGF [10]. In addition to ACE, our lab also identified members of the Wnt pathway that are regulated by VEGF and anti-VEGF [11]. Activating the Wnt pathway using LiCl attenuates VEGF-induced permeability in HRECs, as does overexpression of beta-catenin, a gene that is upregulated by VEGF [11]. While manipulation of ACE and Wnt signaling limits VEGF-induced permeability, it is not abolished. This suggests that VEGF regulates expression of other genes that contribute to permeability.

The goal of this project was to identify additional genes that mediate VEGF-induced permeability of HRECs. To achieve this, transcriptomics analysis was performed to identify genes whose expression is changed by VEGF and reversed by anti-VEGF (anti-VEGF DEGs (differentially expressed genes)). This list of anti-VEGF DEGs was filtered for those that are common to both normal glucose and high glucose conditions, and also capable of governing vascular homeostasis. This process identified components of the Notch signaling pathway and angiopoietin 2 (*ANGPT2*) as such effectors. Further analysis of these genes revealed that (1) VEGF uses distinct effectors to initiate and sustain barrier relaxation and (2) in addition to genes that promote barrier relaxation, VEGF induces effectors that limit relaxation.

## 2. Results

### 2.1. Some VEGF and Anti-VEGF-Regulated Genes Were Insensitive to Hyperglycemia

The goal of this study was to identify genes that contribute to VEGF-induced barrier relaxation. To this end, we prioritized VEGF-regulated genes whose expression is returned to the basal level by anti-VEGF, which we termed anti-VEGF differentially expressed genes (DEGs) (Figure 1a). This approach, applied to HRECs cultured in high glucose (HG; 30 mM; an in vitro model of diabetic retinopathy) conditions, identified a number of such genes [10,11]. We extended this strategy to HRECs cultured in normal glucose (NG; 5 mM) conditions because anti-VEGF therapy is effective at restoring barrier integrity not only in patients with diabetes (PDR, DME), but also in patients whose blood glucose is not chronically elevated (nAMD) [12,13].

RNA-Seq analysis indicated that the glucose concentration profoundly influenced gene expression of confluent cultures of primary human retinal endothelial cells (HRECs). Culturing HRECs in HG conditions alone altered the expression of 2423 genes (Figure 1b). The transcriptional response to VEGF was also altered, as expression of fewer genes changed in NG versus HG conditions; 2613 versus 4372 (Figure 1b). This happened despite VEGFR2 being more abundant in NG cells (Appendix A). Furthermore, a larger percentage of VEGF-regulated genes were reversed by anti-VEGF in NG versus HG; 30.8% (804/2613) versus 6.4% (279/4372) (Figure 1b). Thus, VEGF changed expression of fewer genes and a greater percentage of them were reversed by anti-VEGF in NG versus HG cells. In contrast to the large effect on gene expression, the glucose concentration did not influence permeability in response to a saturating dose of VEGF (kinetics and extent of relaxation); however, it did slightly influence anti-VEGF-mediated re-closure, which was complete and incomplete in HG and NG HRECs, respectively (Appendix A, [10]).

We used the above-described glucose effect to filter the list of anti-VEGF DEGs. Instead of pursuing anti-VEGF DEGs that were unique to either NG or HG cells, we focused on the list of 213 common anti-VEGF DEGs expecting that the most important ones were engaged irrespective of glucose concentration. Ingenuity pathway analysis of these common anti-VEGF DEGs indicated that nine of the top 10 pathways pertained to the cell cycle (Figure 1c). The genes associated with the top 10 pathways are listed in Figure 1d and Appendix A. The experimental system used in these studies involves primary endothelial cells that were grown to confluence and, therefore, had undergone density-dependent inhibition of the cell cycle [14]. Indeed, VEGF and anti-VEGF have minimal effects on the cell cycle under these conditions [10]. Consequently, we did not pursue these in silico results that pointed to the cell cycle.

To further interrogate the list of common anti-VEGF DEGs for those that are most likely to govern permeability, we determined which of them were also on the list of 670 known vascular homeostasis (VH) genes [11]. There were 18 (Figure 2a), which we organized by their associated signaling pathway (Figure 2b). The pathways to which these genes belonged included the Notch pathway, the endothelial inflammatory response pathway, and the senescence pathway (Figure 2b). We chose to pursue the Notch pathway because it harbored the greatest number of gene candidates, in addition to its known role in diabetic retinopathy.

### 2.2. Notch Signaling Was Required for VEGF to Sustain Barrier Relaxation

Clinical and wet lab results published by other investigators supported our in silico findings that activation of the Notch pathway was associated with VEGF/anti-VEGF control of the endothelial barrier. Activation of the Notch pathway induces vessel leakage in murine retinas, and some of the Notch ligands (delta-like ligand 4 (DLL4) and Jagged1) are elevated in patients with diabetic macular edema [15,16]. To determine whether Notch signaling was required for VEGF-induced permeability, we tested if a global Notch pathway inhibitor (DAPT) affected basal and VEGF-induced barrier relaxation. In this series of experiments, we chose to pharmacologically instead of molecularly antagonize the Notch pathway because the pharmacological approach allowed us to inhibit Notch either before, or after addition of VEGF. Inhibiting the Notch pathway compromised basal barrier function (Figure 3a). This effect did not appear to be the result of DAPT toxicity, because DAPT had no effect on the appearance or density of the cells, and the concentration used (10 uM) was below what other groups report using [17]. Furthermore, DAPT-treated cells were still responsive to VEGF (Figure 3). However, blocking the Notch pathway after the barrier had been relaxed with VEGF attenuated the extent to which the barrier remained open in the continued presence of VEGF (Figure 3a,b). Together, these observations suggest that the Notch pathway contributes to regulating permeability in two ways; by enforcing the basal barrier, and by enabling a VEGF-relaxed barrier to stay maximally open.

We also investigated if DAPT suppressed VEGF-induced relaxation of the barrier. We altered the design of the experiment shown in Figure 3a to add DAPT prior to VEGF, instead of after VEGF had relaxed the barrier. Surprisingly, VEGF-induced relaxation was unaffected by suppressing the Notch pathway (Figure 3c,d). Similar behavior was observed in HG HRECs (Appendix A). These data indicate that while Notch contributed to keeping the barrier open, it was dispensable for the VEGF-driven transition from a closed to open barrier.

To further investigate this phenomenon, we sought to determine whether the timing of VEGF-dependent changes in expression of Notch pathway genes corresponded to initiating or sustaining barrier relaxation. We observed that VEGF upregulated expression of *NOTCH4*, *HES4*, *PSEN2*, and *NRARP* at 20 h, but not at 6 h (Figure 4a,d). Furthermore, *HEY1* and *DLL4* were not fully induced by VEGF until 20 h (Figure 4e,f). The 6 and 20 h time points correspond to VEGF-driven barrier opening and persistent relaxation, respectively (Appendix A and [10]). Similar results were observed in NG and HG cells (Figure 4). Taken together, these data indicate that VEGF-dependent changes in expression of Notch pathway genes corresponded to sustaining VEGF-induced barrier relaxation instead of initiating it.

### 2.3. VEGF and Anti-VEGF Regulated ANGPT2 Expression

The existence of VEGF effector genes (members of the Notch pathway) that sustain barrier relaxation, but do not initiate it, suggest that VEGF-induced relaxation involves distinct phases that depend on distinct effectors. Our previous investigation of the Wnt pathway’s contribution to VEGF-driven permeability supports this concept. VEGF altered expression of some members of the Wnt pathway within 0.5 h (coincident with barrier relaxation), whereas expression of others changed only after the barrier was fully relaxed (8 h post VEGF) [11]. This suggests that certain VEGF-regulated genes are important for initiating barrier relaxation, while others impact barrier integrity after an initial relaxation phase.

To identify which of the genes listed in Figure 2b are important for initiating barrier relaxation, we performed RNA-Seq analysis on HG HRECs treated with VEGF or VEGF + anti-VEGF for the minimum time needed for barrier relaxation or reclosure, respectively (Figure 5a). Using this dataset, we determined that *ANGPT2*, *DLL4*, *HEY1*, *VCAM1*, and *PDGFB* were altered by VEGF at 6 h, and then reversed by anti-VEGF at 20 h (Figure 5b). Of these five genes, *VCAM1* and *PDGFB* require interactions with other cell types (leukocytes and mesenchymal cells, respectively) and were not studied further in our HREC monoculture system [18,19]. The experiments shown in Figure 3c demonstrated that the Notch pathway was not required for VEGF-driven relaxation of the barrier; hence, *DLL4* and *HEY1* were not further pursued. *ANGPT2* was upregulated by VEGF at both the 6 h and 20 h timepoints, and this upregulation was reversed by anti-VEGF. qRT-PCR analysis confirmed the RNA-Seq results (Figure 5c). Consequently, we further investigated *ANGPT2′*s contribution to the VEGF-induced transition from a closed to an open barrier.

### 2.4. Ang2 Limited the Extent of VEGF-Induced Permeability

Clinical and wet lab results published by other investigators supported our in silico findings that angiopoietin 2 (Ang2; gene product of *ANGPT2*) contributes to VEGF/anti-VEGF control of the endothelial barrier. Ang2 binds to the Tie2 receptor, preventing Ang1 from binding and promoting vascular stability [20]. Additionally, Ang2 has been reported to be elevated in the vitreous fluid of diabetics [21]. To determine if Ang2 contributed to VEGF-induced relaxation of the barrier, we determined if suppressing expression of endogenous *ANGPT2* affected VEGF-induced permeability. The siRNA-mediated approach reduced expression of *ANGPT2* by 77% (Appendix A). While basal barrier function was unaffected, the extent to which VEGF relaxed the barrier increased in *ANGPT2*-silenced cells (Figure 6a,b; Appendix A). These results demonstrate that VEGF-induced expression of Ang2 did indeed contribute to barrier relaxation; however, instead of promoting relaxation, Ang2 suppressed it. We conclude that VEGF induces expression of genes that not only promote barrier relaxation, but also limit it.

## 3. Discussion

We applied a series of filters to our transcriptomic dataset to identify molecular effectors of VEGF-induced permeability. Our efforts identified not only genes that promote relaxation of the endothelial barrier (members of the Notch pathway), but also genes that limit VEGF-induced permeability (*ANGPT2*). Finally, we found that VEGF relies on distinct genes to initiate and sustain a breached barrier.

We observed that while Notch signaling was required for VEGF to sustain barrier relaxation, it was dispensable for the transition from closed to open (Figure 3a,d). To our knowledge, this represents the first report of VEGF-regulating transcription of a pathway which acts to specifically sustain barrier relaxation and establishes distinct classes of VEGF-induced genes. The first class includes genes like *ACE* that act as effectors that initiate barrier opening [10]. The second class includes genes like members of the Notch pathway that sustain relaxation of a breached barrier. While the mechanism of sustaining an open barrier requires further investigation, the discovery of a distinct class of VEGF-induced genes provides a therapeutic opportunity to target effectors that contribute to chronic vascular leakage in the retina. Our discovery that Notch signaling acts to sustain VEGF-induced barrier relaxation in primary human retinal endothelial cells, when coupled with the elevated Notch ligands DLL4 and Jagged1 observed in patients with diabetic macular edema [15], suggests that therapeutics modulating Notch signaling could help suppress VEGF-mediated vascular leakage.

In contrast to our observations, reports from other groups were the basis for our expectation that knockdown of *ANGPT2* would reduce the extent of barrier relaxation in response to VEGF. For instance, Ang2 is known to interact with the Tie2 receptor, preventing its phosphorylation in response to angiopoietin 1 (Ang1) [22]. This competitive inhibition of Tie2 phosphorylation by Ang2 removes the vessel-stabilizing effects of Ang1, leading to increased endothelial permeability [23,24,25], although there have been reports of Ang2 promoting Tie-2 phosphorylation and vessel stability in certain conditions [26]. Surprisingly, knockdown of endogenous *ANGPT2* allowed VEGF to relax the barrier to a greater extent. We observed that Tie2 is minimally phosphorylated in HRECs, even in the presence of Ang1 and VE-PTP inhibition (Appendix A). The weak pTie2 signal that we observed in cultured HRECs has also been reported when using other types of cultured endothelial cells [27]. In our in vitro system, Ang2 is, therefore, unlikely to alter barrier function by preventing phosphorylation of Tie2. Instead, Ang2 may engage Tie2-independent signaling, of which Ang2 has been shown to bind directly to α_v_β_3_, α_v_β_5_, and α_5_β_1_ integrins [28,29]. Ang2-integrin interactions could promote junctional stability, and account for the increased VEGF-induced relaxation observed when *ANGPT2* was knocked down in this system. Ang2 signaling or the balance of Ang1/Ang2 signaling may also be important for the recovery phase in response to inflammatory stimuli [26,30], and, thus, account for the barrier stabilizing effect of Ang2 observed here.

While *ANGPT2* knockdown experiments provided proof of principle that endogenous Ang2 prevents maximal barrier relaxation in response to VEGF, the effect was modest. The effect’s magnitude may be due to a number of factors including the amount of endogenous Ang2 present in the system, and/or the time needed to deplete the endogenous Ang2. Ang2 is stored in Weibel-Palade bodies in endothelial cells before being released [20]. Therefore, while *ANGPT2* message is depleted by the siRNA, the protein may continue to be released and continue limiting VEGF-induced relaxation. Additionally, the extent of knockdown was not complete (Appendix A), which could have limited the effect size and contributed to the variable effect size and timing observed in replicate experiments (Appendix A). While these factors may have reduced the consequences of *ANGPT2* knockdown in our experimental setting, the importance of endogenous Ang2 may differ in other scenarios. Furthermore, the in vitro setting of these experiments may not completely model the complex biology involved in diabetic retinopathy.

In this study, we used glucose concentration as a filter that identified Notch signaling and Ang2 as molecular effectors that affect HREC barrier relaxation in response to VEGF. Our previous studies utilized the transcriptomes of hyperglycemic HRECs stimulated with VEGF with and without anti-VEGF to identify ACE and the Wnt pathway as molecular effectors that promote or limit VEGF-induced barrier relaxation, respectively [10,11]. While this previous approach provided insights into the transcriptional component of VEGF-induced barrier relaxation, it yielded a total of 279 anti-VEGF DEGs (Figure 1b). Therefore, additional filters were needed to distill this list down to the effectors that are most relevant to VEGF-induced barrier relaxation. Here, we filtered this dataset against the anti-VEGF DEGs obtained from euglycemic HRECs, then focused on genes associated with vascular homeostasis, which distilled this list down to 18 genes (Figure 2). From this list of 18 genes, we identified that the Notch components *NOTCH4*, *HES4*, *NRARP*, *DLL4*, and *HEY1* are induced by VEGF and act as effectors that sustain barrier relaxation. Further filtering of this list led to the finding that *ANGPT2* induction limits VEGF-induced barrier relaxation in HRECs.

While not the focus of the study, it is noteworthy that hyperglycemia had a profound effect on the molecular landscape of HRECs. Culturing HRECs in HG conditions for >10 days changed the basal expression of 2423 genes, which is comparable to the number of genes that changed when NG were exposed to VEGF (2613). This sizable transcriptomic shift suggests that exposing endothelial cells to hyperglycemia induces epigenetic changes that transform the chromatin landscape and alter the transcriptional response to stimuli, including VEGF. Indeed, VEGF altered expression of 67% more genes in HG versus NG HRECs, and a smaller percentage (6.4% vs. 30.8%) of VEGF-regulated genes were reversed by anti-VEGF in HG versus NG conditions. Even though expression of thousands of genes were different between NG and HG HRECs, barrier relaxation was indistinguishable in response to a saturating dose of VEGF. This result suggests that VEGF induces many transcriptional changes in HG HRECs that do not impact barrier integrity, which underscores the power of using glucose as a transcriptomic filter. While the substantial transcriptional shift induced by hyperglycemia did not perturb permeability driven by a high concentration of VEGF, it may contribute to other facets of diabetes-associated pathologies.

## 4. Materials and Methods

### 4.1. Culture of Human Retinal Endothelial Cells

Primary human retinal endothelial cells (HRECs) were purchased from Cell Systems (Kirkland, WA, USA). They were derived from donor A, a 26-year-old Caucasian male. HRECs were cultured in Lonza endothelial cell basal medium-2 (EBM-2, CC3156) supplemented with Lonza SingleQuots endothelial cell growth medium-2MV (EGM-2V, CC4147) from Lonza Bioscience (Verviers, Belgium). The concentration of VEGF in complete Lonza media (EBM2 + EGM-2V) was 2 ng/mL. Normal-glucose HRECs were cultured in this medium (5 mM) with no additional glucose added. To establish an in vitro model of diabetic retinopathy, HRECs were cultured in high-glucose medium containing 30 mM glucose for at least 10 days; medium was changed every 24 h.

### 4.2. RNA Sequencing

For the first sequencing experiment, HRECs were cultured in normal- or high-glucose medium and treated with vehicle (PBS) or 1 nM recombinant human VEGF-A (VEGF 165; 293-VE, R&D Systems, Minneapolis, MN, USA) for 48 h, or VEGF for 24 h then 500 nM aflibercept (Eylea, Regeneron Pharmaceuticals Inc., Tarrytown, NY, USA) for 24 h. For the second sequencing experiment, high-glucose HRECs were treated with vehicle (PBS), 2 nM recombinant human VEGF-A (VEGF 165, 100-20, Peprotech, Inc, Rocky Hill, NJ, USA) for 6 h, VEGF for 20 h, or 6 h VEGF + 14 h 1 uM anti-VEGF (aflibercept). Total RNA was extracted from the HRECs using the RNeasy Mini Kit (Qiagen). Quality was assessed using Tapestation by the UIC Genome Research Core. Libraries were prepared and sequenced by the University of Chicago Genomics Facility. The sequencing for the first sequencing experiment has been described previously [10], the sequencing for the second sequencing experiment was performed on the Illumina NovaSeq 6000 (Illumina, San Diego, CA, USA). featureCounts for the second sequencing experiment are available in Appendix A.

### 4.3. RNA-Seq QC and Quantification

Basic processing of the raw data was performed by the University of Illinois at Chicago Research Informatics Core. Data processing consisted of general quality-control using FastQC [31] before reads were aligned to the v.hg38 human reference genome in a splice-aware manner using STAR [32]. The abundance of genomic features (e.g., genes) were quantified as raw counts based on read alignments using featureCounts [33]. Gene isoforms were quantified using Kallisto, using k-mer based pseudo-alignment and expectation maximization to probabilistically assign reads to isoforms [34].

Differential statistics were then performed on the mapped featureCounts using the edgeR software package version 4.1.1 [35,36]. Differentially expressed genes (DEGs) were identified as genes which had a false discovery rate of 5% (0.05) between groups, as determined by the Benjamini-Hochberg procedure [37]. Genes were determined to be reversed by anti-VEGF in the first sequencing experiment if they were (1) DEGs between the VEGF and anti-VEGF groups and/or (2) the absolute value of the fold change/control was lower in the anti-VEGF group than the VEGF group. Genes were determined to be reversed by anti-VEGF in the second sequencing experiment if they were (1) DEGs between 20 h VEGF and anti-VEGF groups and/or (2) the absolute value of the fold change/control was lower in the anti-VEGF group than the 6 h VEGF group.

### 4.4. Western Blotting

HRECs were plated at full confluency two days prior to lysate harvesting. The cells were rinsed in ice-cold phosphate-buffered saline and lysed in 4% sodium dodecyl sulfate and 5.6 mol/L 2-mercaptoethanol. Lysates were denatured by boiling in sample buffer (10 mmol/L EDTA; 20% glycerol; 200 mmol/L Tris-HCl, pH 6.8; and 0.2% bromophenol blue). Proteins were resolved on a 7.5% SDS-polyacrylamide gel and subjected to Western blot analysis. For the Tie2 phosphorylation experiments, HRECs were treated with 50 ng/mL recombinant human angiopoietin 1 (Peprotech 130-06, Rocky Hill, NJ, USA) and 10 uM Razuprotafib (AKB-9778; Medkoo Biosciences 329509, Morrisville, NC, USA). Antibodies used for immunoblotting were VEGFR2 (CST 2479S, Danvers, MA, USA), b-actin (CST 4970S), phospho-Tie2 (CST 4221S), and Tie2 (CST 7403S). Protein abundance was quantified using ImageJ (NIH, Bethesda, MD, USA).

### 4.5. Electric Cell-Substrate Impedance Sensing (ECIS) Assay

Cell permeability was determined by electric cell-substrate impedance sensing (ECIS) assay. Measurements were made using an ECIS ZTheta instrument (Applied Biophysics, Troy, NY, USA) housed within a standard tissue culture incubator, as described previously [10,11]. Briefly, HRECs were plated at full confluency into the 8-well chamber slides equipped with gold-coated microelectrodes (Applied Biophysics #8W10E+) two days prior to recording. For the Notch inhibition experiments, DAPT (N-[N-(3,5-difluorophenacetyl)-l-alanyl]-s-phenylglycinet-butyl ester, D5942) was purchased from Sigma-Aldrich (St. Louis, MO, USA). DAPT was used at a concentration of 10 uM and added either 1 h before or 10 h after stimulation with 2 nM recombinant human VEGF-A (VEGF 165, 100-20, Peprotech, Inc, Rocky Hill, NJ, USA). For the Ang2 knockdown experiments, HRECs were transfected with 10–15 nM ON-TARGETplus SMARTPool Angpt2 or non-targeting pool siRNA (Horizon Discovery, Waterbeach, UK) for 24 h with simultaneous VEGF stimulation. After 24 h, the transfection medium was replaced with fresh medium containing VEGF.

### 4.6. RNA Extraction and Real-Time PCR

Total RNA was isolated from HRECs using the Qiagen RNeasy mini kit (Qiagen, Hilden, Germany). cDNA was generated from 2 ug of total RNA using the High-Capacity cDNA Reverse Transcription Kit (Invitrogen, Carlsbad, CA) using random hexamer primers, following the manufacturer’s instructions. The resulting cDNA was diluted to a concentration of 10 ng/uL and 1.67 uL was aliquoted for each 6.67 uL qPCR reaction (SYBR Green, Bio-Rad, Hercules, CA, USA). Reactions were performed using primers (Integrated DNA Technologies, Coralville, IA, USA) at a concentration of 300 nM each. Technical replicates of PCR reactions were run and quantitated using the Quantstudio 7 Flex Real-Time PCR system (Applied Biosystems, Foster City, CA, USA). Results were normalized to glyceraldehyde 3-phosphate dehydrogenase (*GAPDH*) expression and expressed as arbitrary units. Primer sequences are listed in Appendix A.

### 4.7. Quantification and Statistical Analysis

All data are presented as the mean ± standard deviation (SD) and were analyzed by unpaired two-tailed Student’s *t*-test, with an α of 0.05. Graphs were created using GraphPad Prism version 9.5.0 (GraphPad Software, San Diego, CA, USA).

## 5. Conclusions

The paracellular barrier between retinal endothelial cells relaxes in response to molecular effectors that are induced by VEGF. This phenomenon occurs in euglycemic and hyperglycemic conditions, despite drastic differences in the volume of effectors regulated by VEGF and anti-VEGF under these conditions. Of the effectors that are regulated under both glucose conditions, there exists a subset which serves to sustain, but do not initiate opening of the barrier. This subset of effectors, including the Notch pathway, could be prioritized to mitigate existing leakage of retinal vessels.

## Figures and Tables

**Figure 1 ijms-24-06402-f001:**
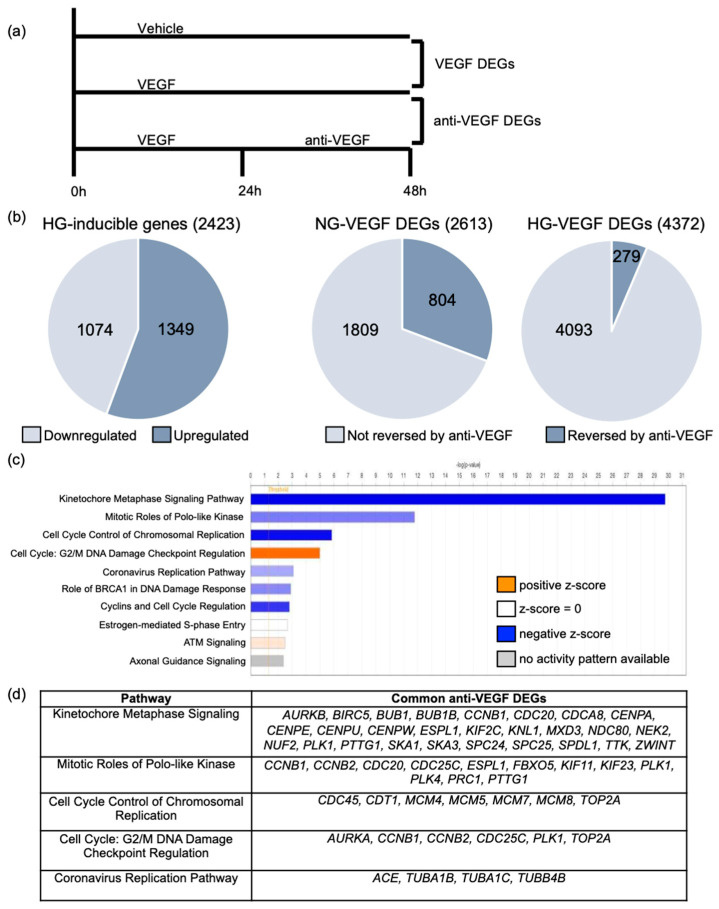
The glucose concentration changed both the number of genes that responded to VEGF and the percentage of them that were reversed by anti-VEGF. (**a**) Schematic depicting treatment conditions of normal-(NG) and high-glucose (HG) human retinal endothelial cells (HRECs) analyzed using RNA-Seq. HRECs were treated with vehicle (PBS), vascular endothelial growth factor (VEGF; 1 nM), or anti-VEGF (500 nM) for times listed. (**b**) Pie charts representing the number of genes differentially expressed in NG and HG HRECs in response to VEGF. VEGF DEGs were further organized by their responsiveness to anti-VEGF. (**c**) List of the top 10 pathways regulated by VEGF/anti-VEGF in NG and HG HRECs, as determined by ingenuity pathway analysis. Pathways were sorted by statistical significance. (**d**) Chart showing the genes associated with the five top pathways regulated by VEGF/anti-VEGF in both glucose concentrations.

**Figure 2 ijms-24-06402-f002:**
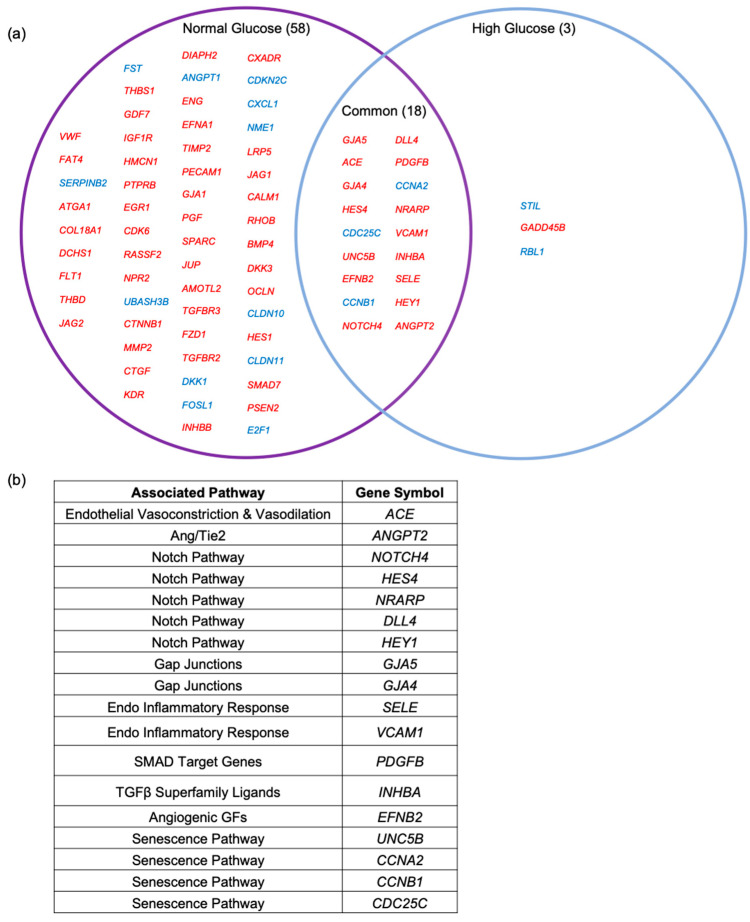
Eighteen of the common anti-VEGF DEGs were known to govern vascular homeostasis. (**a**) Venn diagram depicting the list of VEGF/anti-VEGF regulated genes that are known vascular homeostasis (VH) genes in normal- and high-glucose HRECs. Genes listed in red were upregulated in VEGF-treated cells; genes listed in blue were downregulated in VEGF-treated cells. (**b**) Chart showing the pathways associated with the 18 VH genes that are regulated by VEGF/anti-VEGF in both glucose concentrations.

**Figure 3 ijms-24-06402-f003:**
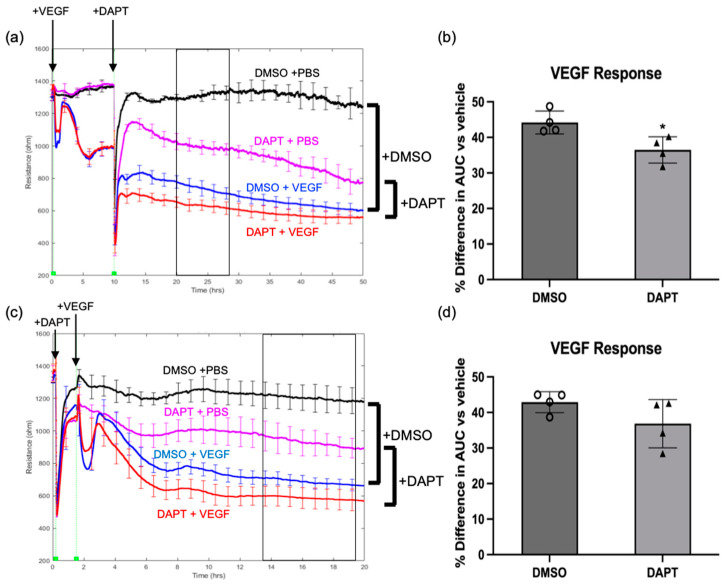
Notch signaling was required for VEGF to sustain barrier relaxation. (**a**) Electric cell-substrate impedance sensing (ECIS) of confluent monolayers of normal-glucose HRECs was measured briefly before the addition of VEGF (2 nM) or vehicle (PBS) and the resistance is presented in ohms. Resistance was measured for 10 h before the addition of DAPT (10 uM) or vehicle (DMSO) at time indicated by arrows. The box represents the time period used for area under curve (AUC) calculation. (**b**) The percent difference in the resistance AUC for HRECs stimulated with VEGF, then given DAPT or vehicle (DMSO) 10 h later. Percent difference was calculated relative to HRECs that received VEGF vehicle (PBS), then DAPT or DMSO 10 h later. AUC was calculated for the period 12–18h after DAPT was added. While the effect of DAPT on extent of barrier relaxation during the 12–18 h time period was not statistically significant in two other experiments, these replicate experiments showed the same trend. Importantly, the effect of DAPT was statistically significant in all 3 independent experiments when a longer time period (10–40 h) was considered. (**c**) ECIS of confluent monolayers of normal-glucose HRECs was measured briefly before the addition of DAPT (10 uM) or vehicle (DMSO) at time indicated by arrows. After 1 h, cells received VEGF (2 nM) or vehicle (PBS), the resistance was measured is presented in ohms. The box represents the time period used for AUC calculation. (**d**) The percent difference in the resistance AUC for HRECs pretreated with DAPT or vehicle (DMSO), then stimulated with VEGF 1 h later. Percent difference was calculated relative to HRECs that received DAPT or DMSO, then VEGF vehicle (PBS) 1 h later. AUC was calculated for the period 12–18 h after VEGF was added. Similar results are observed in three independent experiments. Data are expressed as mean ± SD. * *p* < 0.05 vs. PBS. n = 4 wells per group.

**Figure 4 ijms-24-06402-f004:**
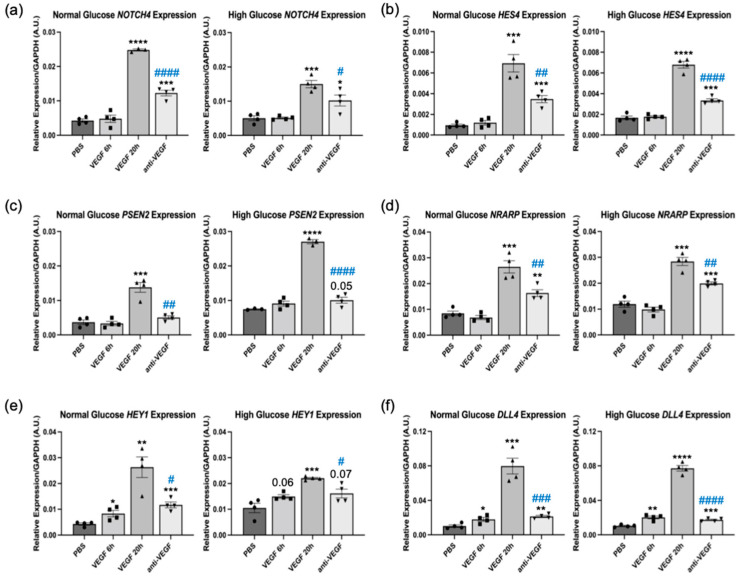
The maximal Notch response to VEGF occurred after 20 h. (**a**–**f**) Quantitative PCR for the expression of (**a**) *NOTCH4*, (**b**) *HES4* (**c**) *PSEN2*, (**d**) *NRARP* (**e**) *HEY1*, and (**f**) *DLL4* in normal- and high-glucose HRECs treated with vehicle, 6 h VEGF, 20 h VEGF, or 6 h VEGF + 14 h anti-VEGF. Expression of genes of interest was normalized to *GAPDH* and presented as relative expression. Similar results are observed in three independent experiments. Data are expressed as mean ± SD. The difference in means between two groups was analyzed using Student’s *t*-test where * *p* < 0.05, ** *p* < 0.01, *** *p* < 0.001, and **** *p* < 0.0001 vs. PBS; # *p* < 0.05, ## *p* < 0.01, ### *p* < 0.001, #### *p* < 0.0001 vs. 20 h VEGF. n = 4 wells per group.

**Figure 5 ijms-24-06402-f005:**
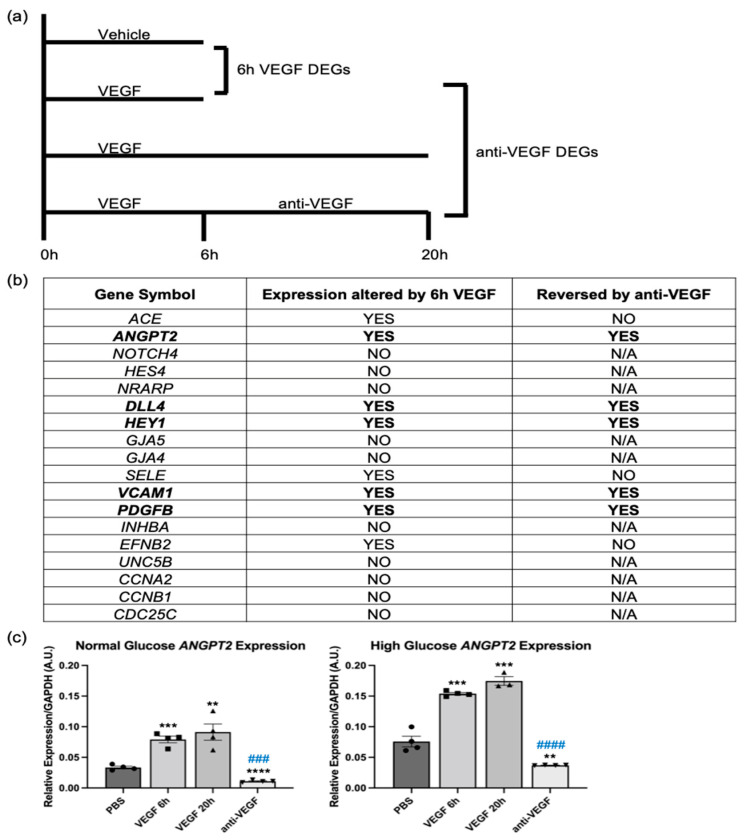
VEGF and anti-VEGF regulated *ANGPT2* expression in both euglycemic and hyperglycemic conditions. (**a**) Schematic depicting the treatment conditions of high-glucose HRECs analyzed using RNA-Seq. HRECs were treated with vehicle (PBS), VEGF (2 nM), or anti-VEGF (1 uM) for times listed. (**b**) Chart showing short-term VEGF/anti-VEGF regulation of the 18 vascular homeostasis genes regulated by long-term VEGF/anti-VEGF exposure. (**c**) Quantitative PCR for the expression of *ANGPT2* message in normal and high-glucose HRECs treated with vehicle, 6 h VEGF, 20 h VEGF, or 6 h VEGF + 14 h anti-VEGF. *ANGPT2* expression was normalized to *GAPDH* and presented as relative expression. Similar results are observed in three independent experiments. Data are expressed as mean ± SD. The difference in means between two groups was analyzed using Student’s *t*-test where ** *p* < 0.01, *** *p* < 0.001, and **** *p* < 0.0001 and ### *p* < 0.001, #### *p* < 0.0001 vs. 20 h VEGF. n = 4 wells per group.

**Figure 6 ijms-24-06402-f006:**
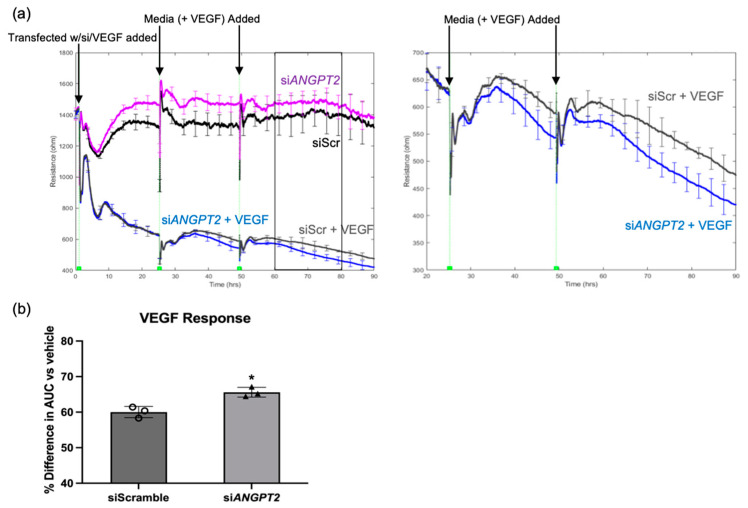
Ang2 limited the extent of VEGF-induced permeability. (**a**) Electric cell-substrate impedance sensing (ECIS) of confluent monolayers of normal-glucose HRECs was measured briefly before transfection with siScramble or si*ANGPT2* (10–15 nM) with simultaneous treatment with VEGF (2 nM) or vehicle (PBS). Resistance is presented in ohms. Transfection media was removed at 24 h and replaced with fresh media containing VEGF or vehicle. The box represents the time period used for area under curve (AUC) calculation. (**b**) The percent difference in the resistance AUC for HRECs transfected with siScramble or si*ANGPT2* and simultaneous VEGF stimulation. Percent difference was calculated relative to HRECs that received VEGF vehicle (PBS) with respective siRNA. AUC was calculated for the period 60–80 h after transfection and VEGF stimulation. Similar results are observed in at least three independent experiments. Data are expressed as mean ± SD. * *p* < 0.05 vs. PBS. n = 3 to 4 wells per group.

## Data Availability

The data presented in this study are available upon request from the corresponding author.

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
