# Peer review of "VEGF Induces Expression of Genes That Either Promote or Limit Relaxation of the Retinal Endothelial Barrier"

_ijms, 2023, doi:10.3390/ijms24076402_

Round 1

Reviewer 1 Report

The authors propose a contribution of the notch pathway in the maintenance of VEGF induced permeability but not its initiation.

While the expression data are overall convincing, the permeability experiments and their interpretation needs clarification.

Figure 3 and corresponding text

-      The effect of DAPT on basal permeability measured by resistance is much higher than that on VEGF induced permeability in both settings (Fig 3a and 3c). The authors should discuss the extend of compromised basal barrier function vs. attenuation of VEGF-induced permeability. Toxic effects can also cause a decrease of resistance in cell culture. Can the authors exclude a toxic effect of DAPT?

-      The authors compare a 30 h covering AUC for the DAPT after VEGF design (Fig 3a and 3b) with a 3 h covering AUC for the DAPT before VEGF design (Figure 3c and 3d) and come to the conclusion that in the later one, VEGF-induced relaxation was unaffected by DAP. Due to the overall only very weak effect of DAPT on VEGF induced permeability this artificial comparison of two completely different AUC ranges is not acceptable. Even if finally, a statistical significance can be calculated using comparable time ranges, what is the relevance of this? The curves look quite identical.

Figure 6 and corresponding text:

The differences between siScr and siANPGT2 on vehicle and VEGF incubated cells is quite low and not at all a strong support for the conclusions the authors made (suppression effect of Ang2 on VEGV-induced permeability). Is there really a biological meaningful effect of the siRNA on VEGF-induced permeability or isn’t it just an irrelevant statistical significance? The authors stated that similar results as depicted in Fig 6 were obtained in at least 3 independent experiments. The statistical power and thus the support of their conclusions, would benefit from a combined evaluation of all performed experiments.

Author Response

The authors propose a contribution of the notch pathway in the maintenance of VEGF induced permeability but not its initiation.

While the expression data are overall convincing, the permeability experiments and their interpretation needs clarification.

 Figure 3 and corresponding text

-      The effect of DAPT on basal permeability measured by resistance is much higher than that on VEGF induced permeability in both settings (Fig 3a and 3c). The authors should discuss the extend of compromised basal barrier function vs. attenuation of VEGF-induced permeability. Toxic effects can also cause a decrease of resistance in cell culture. Can the authors exclude a toxic effect of DAPT?

The Results section related to Figure 3 has been re-written to address these points. 

-      The authors compare a 30 h covering AUC for the DAPT after VEGF design (Fig 3a and 3b) with a 3 h covering AUC for the DAPT before VEGF design (Figure 3c and 3d) and come to the conclusion that in the later one, VEGF-induced relaxation was unaffected by DAP. Due to the overall only very weak effect of DAPT on VEGF induced permeability this artificial comparison of two completely different AUC ranges is not acceptable. Even if finally, a statistical significance can be calculated using comparable time ranges, what is the relevance of this? The curves look quite identical.

We revised the presentation of Fig 3 to address these issues. The changes include comparing the same time interval in Fig 3a/b and 3c/d; altering the bar graphs in Fig 3b and 3d accordingly; and adding brackets to the TEER tracing in Fig 3a and c to improve the clarity of the experimental design of the experiment.  Finally, we expanded the legend of Fig 3 by including additional details regarding replicate experiments. 

Figure 6 and corresponding text:

The differences between siScr and siANPGT2 on vehicle and VEGF incubated cells is quite low and not at all a strong support for the conclusions the authors made (suppression effect of Ang2 on VEGV-induced permeability). Is there really a biological meaningful effect of the siRNA on VEGF-induced permeability or isn’t it just an irrelevant statistical significance? The authors stated that similar results as depicted in Fig 6 were obtained in at least 3 independent experiments. The statistical power and thus the support of their conclusions, would benefit from a combined evaluation of all performed experiments.

Replicate experiments have been added as supplemental Figure S6 to support this conclusion. Furthermore, we expanded the consideration of the size of the Ang2 knock down effect in the Discussion section.

Reviewer 2 Report

In this paper, McCann et al., describes that Notch signaling genes are up-regulated to maintain VEGF-mediated vascular relaxation after VEGF stimulation, while Ang2 is also up-regulated to counteract to this response at same time. In general, VEGF initiates barrier breaking by phosphorylation of VE-cadherin by direct interaction of phosphorylated VEGFR2 and then Dll4 also phosphorylates VE-cadherin after VEGF induces its expression. Ang2 is usually functioned as a barrier destabilizer, but the balance of Ang1/Ang2 or Tie-2 independent Ang2 signaling may be important in the recovery phase from inflammatory stimuli. Due to the poor activation status of Tie-2 in response to Ang1 in cells the authors used, Ang2 may mainly activate Tie-2 independent signaling as the authors discussed, but not demonstrated. And it may be possible that the phenomenon the authors observed may not be reproducible in well-established endothelial culture or in vivo. This study is well-designed from RNAseq data and resulted gene changes are interesting, but VEGF-induced up-regulation of Notch-signaling genes were already well-studied and the counteracting mechanism of Ang2 was poorly evaluated. And some experimental issues must be addressed.

1.     The effect of DAPT on the basal barrier is too strong to calculate % difference in AUC vs vehicle [=AUC(VEGF)/AUC(PBS)]. Why don’t the authors perform experiments using siRNA against Notch1, Notch4 or Dll4? Especially, Notch4 and Dll4 were up-regulated 20 hours, but not 6 hours, after the VEGF treatment.

2.     The RNAseq data should be provided as a supplementary table.

3.     The unit of TEER should be ·cm2. I think the authors measured electrical resistances using an incubator-type measuring device (the method was not described) but I doubt that theses resistances were subtracted by those of blank wells.  

4.     In Figures 4 and 5c, static analysis should be performed by ANOVA.

5.     For endothelial culturing, complete EGM-2MV medium contains approx. 1 nM VEGF-A. The authors should describe that EGM2-MV contained all supplements except for VEGF. And why did the authors use two different VEGF (one is from R&D system and the other is from Peprotech)?    

6.     In line 197, Table 2b may be Figure 2b.

Author Response

In this paper, McCann et al., describes that Notch signaling genes are up-regulated to maintain VEGF-mediated vascular relaxation after VEGF stimulation, while Ang2 is also up-regulated to counteract to this response at same time. In general, VEGF initiates barrier breaking by phosphorylation of VE-cadherin by direct interaction of phosphorylated VEGFR2 and then Dll4 also phosphorylates VE-cadherin after VEGF induces its expression. Ang2 is usually functioned as a barrier destabilizer, but the balance of Ang1/Ang2 or Tie-2 independent Ang2 signaling may be important in the recovery phase from inflammatory stimuli. Due to the poor activation status of Tie-2 in response to Ang1 in cells the authors used, Ang2 may mainly activate Tie-2 independent signaling as the authors discussed, but not demonstrated. And it may be possible that the phenomenon the authors observed may not be reproducible in well-established endothelial culture or in vivo. This study is well-designed from RNAseq data and resulted gene changes are interesting, but VEGF-induced up-regulation of Notch-signaling genes were already well-studied and the counteracting mechanism of Ang2 was poorly evaluated. And some experimental issues must be addressed.

The Discussion has been revised and expanded to include consideration of potential context-dependent effects of Ang2 signaling, including the recovery phase from inflammatory stimuli.

1. The effect of DAPT on the basal barrier is too strong to calculate % difference in AUC vs vehicle [=AUC(VEGF)/AUC(PBS)]. Why don’t the authors perform experiments using siRNA against Notch1, Notch4 or Dll4? Especially, Notch4 and Dll4 were up-regulated 20 hours, but not 6 hours, after the VEGF treatment.

We expanded the Results (related to Fig 3) to include the rationale for the design of this series of experiments in order to explain the importance of the timing of Notch pathway inhibition, which is easier to accomplish with a pharmacological approach.  

2. The RNAseq data should be provided as a supplementary table.

As requested, we have included the RNAseq data as supplementary Table 2.   

3. The unit of TEER should be Ω·cm2. I think the authors measured electrical resistances using an incubator-type measuring device (the method was not described) but I doubt that theses resistances were subtracted by those of blank wells. 

As requested, we indicate that the units of the TEER tracings are ohms (this information is now in the figure legends). In addition, we expanded the Materials & Methods section to provide more experimental detail for how the TEER assays were performed. 

4. In Figures 4 and 5c, static analysis should be performed by ANOVA.

For the qPCR data, the statistical analysis was performed by comparing the difference in means between two groups (i.e. VEGF 6h vs PBS) and therefore t-test was deemed the most appropriate statistical test. The legends for these figures have been revised to indicate that the comparisons were between two groups.

5. For endothelial culturing, complete EGM-2MV medium contains approx. 1 nM VEGF-A. The authors should describe that EGM2-MV contained all supplements except for VEGF. And why did the authors use two different VEGF (one is from R&D system and the other is from Peprotech)?

The Materials & Methods section has been updated to include that the concentration of VEGF in complete medium (EBM2 supplemented with EGM-2V) is 2 ng/ml. VEGF from two different suppliers was used because we were dissatisfied with the customer service at R&D. 

6. In line 197, Table 2b may be Figure 2b.

We have corrected this typo, thank you for catching our error.

Reviewer 3 Report

Author investigated genes that mediate VEGF induced permeability of retinal endothelial cells in vitro, identified distinct set of genes to induce and sustain barrier relaxation. Here are few comments regarding the result.

1. the high Glucose (30mM) condition used in this manuscript  was also applied in previous paper(IOVS, 2021) from the same lab with similar study design, author should discuss the connection or overlap or differences between this two results briefly as it will be useful.

2. this is an in vitro study, which may not well correlate with actual diabetic retinopathy in vivo, author should point out the limitation of this report .

Author Response

Author investigated genes that mediate VEGF induced permeability of retinal endothelial cells in vitro, identified distinct set of genes to induce and sustain barrier relaxation. Here are few comments regarding the result.

1. The high Glucose (30mM) condition used in this manuscript  was also applied in previous paper(IOVS, 2021) from the same lab with similar study design, author should discuss the connection or overlap or differences between this two results briefly as it will be useful.

A paragraph has been added to the discussion section that addresses the connections to the previous papers which focused on hyperglycemic HRECs and the need for additional (glucose) filters to effectively extract molecular effectors from this dataset.

2. This is an in vitro study, which may not well correlate with actual diabetic retinopathy in vivo, author should point out the limitation of this report.

We thank the reviewer for this suggestion and have acknowledged this limitation in the discussion section.

Round 2

Reviewer 2 Report

As a reviewer, I have checked previous author’s paper (https://iovs.arvojournals.org/article.aspx?articleid=2777911), the authors have described that TEER is measured by ECIS Z-theta. But, accurately, ECIS (electric cell-substrate impedance sensing) and TEER (trans-endothelial electrical resistance) are different techniques, although both can represent the barrier strength of endothelial cells. The authors must amend it in this paper.

And the author’s cited reference No. 27 as if they used retinal endothelial cells as well. This is inappropriate and it should be removed.  

And gene names should be provided as well in S. Table 2.

Author Response

As a reviewer, I have checked previous author’s paper (https://iovs.arvojournals.org/article.aspx?articleid=2777911), the authors have described that TEER is measured by ECIS Z-theta. But, accurately, ECIS (electric cell-substrate impedance sensing) and TEER (trans-endothelial electrical resistance) are different techniques, although both can represent the barrier strength of endothelial cells. The authors must amend it in this paper.

We altered the text throughout to replace “TEER” with “ECIS”. We also added a second prior publication in which we described the ECIS technique used with endothelial cells.

And the author’s cited reference No. 27 as if they used retinal endothelial cells as well. This is inappropriate and it should be removed.  

We re-wrote the sentence in which reference 27 was cited in order to clarify the techniques that were used in reference 27.  The new version of this sentence is pasted below: 

“The weak phosphorylation of Tie2 that we observed in cultured HRECs has also been reported when using other types of cultured endothelial cells [27].”

And gene names should be provided as well in S. Table 2.

S.Table 2 has been modified to include gene names.